# Urban Traffic Accident Features Investigation to Improve Urban Transportation Infrastructure Sustainability by Integrating GIS and Data Mining Techniques

**Khanh Giang Le** **, Quang Hoc Tran * and Van Manh Do**

Faculty of Civil Engineering, University of Transport and Communications, No. 3 Cau Giay Street, Lang Thuong Ward, Dong Da District, Hanoi, Vietnam; gianglk@utc.edu.vn (K.G.L.); manhdv@utc.edu.vn (V.M.D.)
* Correspondence: hoctq@utc.edu.vn

**Abstract:** Urban traffic accidents pose significant challenges to the sustainability of transportation infrastructure not only in Vietnam but also all over the world. To decrease the frequency of accidents, it is crucial to analyze accident data to determine the relationship between accidents and causes, especially for serious accidents. This study suggests an integrated approach using Geographic Information System (GIS) and Data Mining methods to investigate the features of urban traffic accidents in Hanoi, Vietnam aiming to solve these challenges and enhance the safety and efficiency of urban transportation. Firstly, the dataset was segmented into homogenous clusters using the two-step cluster method. Secondly, the correlation between causes and traffic accidents was examined on the overall dataset as well as on each cluster using the association rule mining (ARM) technique. Finally, the location of accident groups and high-frequency sites of accidents (hotspots) were determined by using GIS techniques. As a result, a five-cluster model was created, which corresponded to five common accident groupings in Hanoi. Moreover, the results of the study also identified the types of accidents, the main causes, the time, and the surrounding areas corresponding to each accident group. In detail, cluster 5 depicted accidents on streets, provincial, and national roads caused by motorbikes making up the highest percentage within the groups, accounting for 29.2%. Speeding and driving in the wrong lane in the afternoon and at night were the main causes in this cluster ($C_f \geq 0.9$ and $L_t \geq 1.22$). Next, cluster 2 had the second-highest proportion. Cluster 2 presented accidents between a truck/car and a motorbike on national and provincial roads, accounting for 27.8%. Cluster 1 presented accidents between a truck/car and a motorbike on local streets, accounting for 22%. Cluster 3 illustrated accidents between two motorbikes on the country lanes, accounting for 12.3%. Finally, cluster 4 depicted single-vehicle motorbike crashes, with the lowest rate of 8.8%. More importantly, this study also recommended using repeatability criteria for the same type of accidents or causes to determine the location of hotspots. Also, suggestions for improving traffic infrastructure sustainability were proposed. To our knowledge, this is the first time in which these three methods are applied simultaneously for analyzing traffic accidents.

**Keywords:** traffic accident (TA); hotspots; geographic information system (GIS); association rule mining (ARM); clusters; sustainability

## 1. Introduction

Sustainable transportation contributes to an important part in the accomplishment of the majority of the sustainable goals set forth in the United Nations' 2030 Agenda. One of the primary challenges arising from unsustainable transportation is the considerable annual death toll from road traffic accidents [1]. Each day, global road traffic accidents claim the lives of more than 3500 individuals, and the yearly death toll amounts to over 1.25 million people, a figure that has remained relatively unchanged since 2007. Moreover, these accidents result in about 50 million injuries or permanent disabilities annually. Despite

accounting for just 54 percent of the world's vehicles, the majority of traffic fatalities occur in low- and middle-income nations, with 90 percent of such fatalities occurring within their borders [2]. The significance of road safety as a sustainability concern is evident through its incorporation into the 2030 Agenda for Sustainable Development, which sets a target of reducing global road transport deaths and injuries by 50 percent. Furthermore, by 2030, the agenda aims to guarantee that everyone has a chance to utilize sustainable, safe, and cheap transportation networks [1].

During the year 2022, Vietnam experienced a total of 11,450 traffic accidents, which led to 6384 fatalities and caused injuries to 7804 individuals [3]. Today, the road transportation system stands out as one of the most significant risks to human life, given its complex nature comprising various vehicular traffic flows and the prevalence of pedestrians, particularly in urban areas such as Hanoi, Vietnam. Meanwhile, traffic accidents are unpredictable and can happen in many different situations [4]. There are many causes, both subjective and objective, that cause traffic accidents, including surrounding conditions, road design, traffic, vehicle conditions, driver characteristics, etc. [4,5]. Therefore, analyzing accident data is necessary to identify key factors related to traffic accidents in order to make safe driving recommendations and take appropriate preventive measures [6,7].

Many previous studies have shown that statistical models are popular methods of accident analysis because the relationships between traffic accidents and their causes can be clearly defined [8]. Nevertheless, large dataset analysis presents challenges for these methods. More important, each of these methods also has a set of hypotheses. If these hypotheses are not satisfied, the results could not be reliable [9]. Therefore, data mining techniques have become a powerful tool for extracting hidden information from large datasets, especially in the transport sector [10–12]. Numerous data mining techniques, such as clustering, association rule mining (ARM), and classification, are often used for the study of traffic accidents [12].

Indeed, the variety of factors and accident kinds that occur in various contexts makes it difficult to analyze traffic accident datasets [13,14]. Generally, because of the variety of traffic accident data such as it can include both numerical and categorical types, some important relationships may be obscured. Therefore, numerous earlier studies focused on a specific type of traffic accident or created unique models for each type to reduce heterogeneity and accident data is often segmented using expert knowledge. Nevertheless, it is not guaranteed that each group will have a homogenous set of accidents [15]. In fact, cluster-based analysis of traffic accident datasets enables us to uncover more useful information than investigating datasets without cluster techniques [15]. Therefore, cluster analysis techniques are frequently applied as the first vital step to arrange heterogeneous data into homogenous clusters [5,11,15–17].

There are numerous clustering algorithms, each with its own set of benefits and drawbacks. Partitioning methods, like K-Means, K-Medoids, and K-Modes, are simple, efficient, and widely used in various applications. The K-Means and the K-Medoids methods are widely applied numerical clustering algorithms that use Euclidean distance as a distance metric [17]. The K-Medoids is a modified version of the K-Means algorithm, specifically developed to address sensitivity to outlier data. However, both algorithms are unable to process categorical data [16]. The authors of [5] segmented data using the K-Modes method which is a modified version of the K-Means technique for categorical data. Therefore, while the K-Modes cluster technique excels at processing categorical data, it lacks the capability to simultaneously process numerical and categorical data. In addition, the K-Means, the K-Modes, or the K-Medoids clustering methods require identified numbers of clusters, so these algorithms are not applied in many applications [16].

Hierarchical techniques are a type of unsupervised machine-learning algorithm that is used to group data into hierarchical structures or dendrograms. In contrast to partitioning approaches, which need a predetermined number of clusters, hierarchical methods generate a hierarchical representation of the data by continuously merging or dividing clusters depending on the proximity or distance of data points [16]. The hierarchical approach

is restricted to tiny datasets. The latent class clustering (LCC) algorithm was applied to partition the accident dataset into homogeneous clusters [15]. However, the LCC algorithm faces challenges with the dataset containing a huge number of categorical attributes [18]. Diabetes was diagnosed using the balanced iterative reduction and clustering utilizing the hierarchies (BIRCH) method [19]. The BIRCH will not work if the sizes of the data attributes are over 20 [20].

The clustering method based on density has been critical in determining density-based nonlinear form structures. The method based on density that is most frequently employed is called Density-Based Spatial Clustering of Applications with Noise (DBSCAN). This algorithm can detect clusters of arbitrary size and shape. However, it fails in the presence of clusters of changing density and cannot perform well with data that is highly dimensional [16].

The methods mentioned above all lack the capability to simultaneously process numerical and categorical data. The two-step cluster algorithm was developed to solve this weakness of the preceding techniques [21]. However, this technique is still rarely utilized in traffic accident analyses.

After grouping the accident data into homogeneous clusters, it becomes crucial to efficiently analyze and explore the relationships among traffic accidents and their reasons. Furthermore, the drawbacks of the statistical models mentioned above, the interactive associations between two or more variables, and their impact on the severity of injury were not mentioned. To deal with this issue, the association rule mining (ARM) technique stands out as a significant method for investigating these associations. Without predefined assumptions, the ARM technique enables us to investigate attractive rules from a huge dataset [22]. The ARM technique was applied in several studies related to traffic accident analyses [22–24].

The mentioned methods yield non-spatial outcomes, making it difficult to visualize the cluster locations as well as the locations of each accident within those clusters. Accident locations and their features are stored in a Geographic information system (GIS), making it convenient to determine the causes of each accident. Spatial data is essential in the investigation of traffic accidents. A GIS facilitates the collection, storage, manipulation, querying, analysis, and visualization of spatial data. Moreover, a GIS is a potential approach for spatial analysis to determine the accident hotspots. Identifying the locations of high-frequency accidents, or "hotspots", is one of the most crucial tasks in the effort to decrease the number of traffic accidents [25–28]. The locations of traffic accident hotspots can be determined via the locations of accident clusters [13,29]. Subsequent analysis of these hotspots enables traffic administrators to evaluate them more accurately aiming to prevent fatalities in hotspots and to establish suitable preventive solutions aiming to improve road safety [30]. Additionally, drivers can take advantage of this information to avoid these accident-prone areas [31]. Therefore, there is potential for integrating GIS and data mining approaches in many fields, especially in analyzing traffic accidents.

Therefore, this study proposes a methodology. Firstly, the two-step cluster algorithm was applied to segment the data into homogeneous groups. Secondly, the correlation between cause factors and traffic accidents was identified for each cluster as well as for the entire dataset using the association rule mining technique. Thirdly, the locations of clusters and the locations of each accident within each cluster were graphically illustrated on a map by using GIS. Moreover, the locations of traffic accident hotspots were determined based on the repeatability of the same types of accidents or their causes. Currently, there are not many countries that identify the locations of traffic accident hotspots by using repetition criteria of the same types of accidents or their causes. This is the first time, as far as we are aware, that three techniques are utilized simultaneously to analyze traffic accident data. The data on traffic accidents from 2015 to 2017 in Hanoi, the capital of Vietnam, were used as a case study to investigate and test this methodology. The objective of this article was to propose an integrated approach using GIS and data mining techniques to analyze traffic accidents effectively, thereby gaining a better understanding of the accident causes and providing

efficient remedies. This contribution not only enables us to reduce the number of hazardous accidents but also assists traffic managers and urban planners in establishing sustainable urban transportation policies, management, and development. These are the orders of the remainders: A methodology is proposed and illustrated in Section 2. An examination of a case study is included in Section 3. Section 4 provides results and discussions. Section 5 shows conclusions, limitations, suggestions, and future works.

## 2. Methodology

This research proposed an integrated approach using GIS and data mining techniques to analyze traffic accident data effectively, thereby gaining a better understanding of the accident causes and providing efficient remedies. Figure 1 illustrates the integration of GIS technology and data mining techniques in analyzing traffic accident datasets. The following steps were undertaken to implement the proposed methodology:

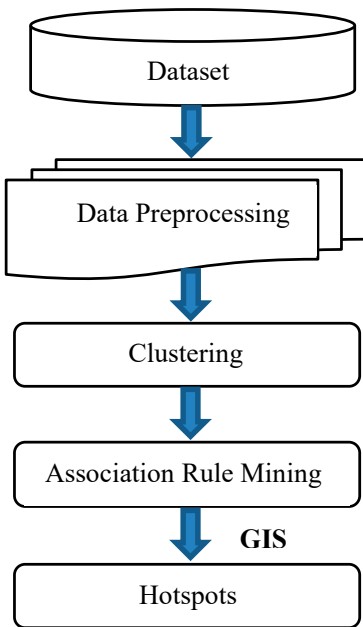

**Figure 1.** The proposed flowchart for analysis.

Firstly, it was necessary to clean the dataset before the analysis process. The process of preparing data makes the dataset accessible for study at a later time. Secondly, the clustering technique used the two-step cluster algorithm to segment the data into homogeneous clusters. Thirdly, the correlation between causative factors and traffic accidents was examined using an association rule mining technique on both the entire dataset as well as on each cluster. Finally, GIS technology was used to locate traffic accident hotspots by using repetition criteria of the same types of accidents or their causes. From there, the appropriate preventive strategies were suggested. It also enables traffic authorities and urban planners to establish suitable policies for developing sustainable urban transportation.

The methods used in this research, including the two-step cluster technique, association rule mining, and kernel density estimation, are explained in more depth.

### 2.1. Data Preparation

In data mining techniques, it is crucial to eliminate noise, handle missing values, and remove unnecessary characteristics. This preprocessing step ensures that the dataset is prepared and ready for subsequent analysis [12,16].

### 2.1.1. Handling Missing Data

The traffic accident dataset in Hanoi is officially recorded by the Hanoi Traffic Police Department. Additionally, relevant parties, including the National Traffic Safety Committee, the National Traffic Police Department, the Hanoi Department of Transport, the Hanoi Department of Health, and press and media agencies, also record this dataset. Consequently, while collecting and preprocessing the data, the authors undertook the extensive task of cross-verifying information from these pertinent sources, despite the considerable time and effort involved. This thorough process contributed to obtaining a nearly comprehensive and accurate dataset.

Specifically, any missing data was supplemented by us from relevant parties. In rare cases where no data was available, the authors proceeded to supplement based on the most probable values.

### 2.1.2. Handling Outliers

For numerical data, outliers can be an unrealistic value or a value that is very different from the rest of the values within that category. In the case of categorical data, outliers may represent unrealistic values, such as an item falling outside the expected range. Additionally, values with an exceptionally low frequency within a data column are also considered potential outliers [16]. In our dataset, which encompasses both numeric and categorical data, the authors handled each type as follows:

1. For numerical data.

The majority of outliers in the traffic accident dataset were derived from data entry errors and natural errors. This research employed the Inter-Quartile Range (IQR) method for their elimination. The IQR provides information about the spread of values in the dataset, calculated as the difference between the upper quartile (75%) and the lower quartile (25%). Outliers were identified as follows:

$$Values < \ Q_1 - 1.5(Q_3 - Q_1) \ \text{Or} \ Values \ > Q_3 + 1.5(Q_3 - Q_1) \tag{1}$$

where $Q_1$ is the lower quartile, $Q_2$ is the median, and $Q_3$ is the upper quartile.

After identifying the outlier values, the authors processed it by replacing the outlier values with another suitable value.

2. For categorical data.

Unlike the case of numeric data, outlier values in categorical data are more difficult to detect. Partly because histograms are difficult to draw, especially when there are many different category values, outliers of this type require expert knowledge of valid values. With categorical data, outliers can occur in one of the following cases:

(1) Due to data entry errors: For example, a part of the data obtained is in uppercase, and another small part is in lowercase, like "car" and "Car", etc. In this case, the authors normalized the values to the same form to remove outliers.

(2) Due to spelling errors, some samples have different values from the rest. To handle misspelled data, the authors drew a histogram showing the frequency of each value in the entire data. Typically, spelling errors were in low-frequency categories. These errors needed to be corrected before going to the next step.

Finally, the authors aggregated the results of outlier handling for both numeric and categorical variables to obtain a comprehensive view of the data. In general, the authors were very passionate about this dataset so the authors reviewed and processed the data set to be the most optimal to ensure the analysis process achieved the most accurate results. After data pre-processing, the final dataset included 18 variables, which were determined satisfactory for the study. The description of the dataset is illustrated in Table 1.

**Table 1.** The variables of traffic accidents.

| Variables | Types | Values |
|---|---|---|
| Vehicle type | Categorical | Bicycle; bus; car; coach; lorry; motorbike; pedestrian; taxi; three-wheeler; tractor; train; truck |
| Accident type | Categorical | Angle; collision with fixed object; head-on; out-of-control; pedestrian-train; pedestrian-vehicle; rear-end; reverse; right angle; sideswipe; turning; vehicle-train |
| Reason | Categorical | Cross the red light; drunk; forbidden road; interchange; not giving way; not paying attention; over-speed; pedestrian crossing; out-of-control; motorcycle carrying 3 people; overtake illegally; puncture; turning illegally; unsafe distance; unsafe reverse; wrong lane |
| Severity index (SI) | Categorical | Moderate; severe; very severe |
| Consequence | Categorical | Fatal; injuries; no injuries |
| Gender | Categorical | Female; male |
| Age | Numerical | 0–15; 16–17; 18–23; 24–29; 30–39; 40–49; 50–59; 60+ |
| Crossroad | Categorical | Crossroad with traffic lights; crossroad with priority road; crossroad with right of way; no crossroad |
| Populated area | Categorical | Yes; no |
| Road type | Categorical | National, provincial road; street; country lane |
| Road sort | Categorical | Single roadway; divided roadway |
| Speed limit | Numerical | 50 km/h; 60 km/h; 70 km/h; 80 km/h; 90 km/h; 120 km/h |
| Surroundings | Categorical | School; hospital; shopping center; recreation center; bus stop; others |
| Weekend | Categorical | No (Monday 1 h–Friday 23 h); Yes (Friday 23 h–Monday 1 h) |
| Hour | Categorical | Morning (6:00 a.m.–11:59 a.m.); afternoon (12:00 p.m.–17:59 p.m.); evening (18:00 p.m.–23:59 p.m.); night (0:00 a.m.–5:59 a.m.) |
| Season | Categorical | Spring; summer; fall; winter |
| Road surface | Categorical | Asphalt; concrete cement |
| No. of victims | Numerical | 0; 1; 2; 3+ |

*2.2. Clustering Analysis*

This study applied the two-step cluster algorithm which was performed in the SPSS statistics software (version 25) because it is appropriate to process large data sets including numerical and categorical data. Moreover, the optimal quantity of clusters was automatically identified [32]. This technique included three steps such as pre-clustering, solving outliers, and clustering. The first step involved scanning the dataset to assess whether the current data could be grouped into existing groups or should initiate a new cluster by using Euclidean and log-likelihood distance standards. Secondly, values that were not suitable for anywhere were treated as outliers and taken away. The third step involved organizing sub-clusters into the ideal quantity of clusters. This step effectively utilized conventional clustering techniques since the quantity of sub-clusters was considerably less than the quantity of original data [21,33].

To determine the number of groups, the indicator AIC (Akaike Information Criterion) or BIC (Bayesian Information Criterion) was calculated for any quantity of groups over a particular interval. Next, an early computation of the number of groups was performed by utilizing that indicator. By identifying the distance's biggest change between the two nearest groups throughout each stage of hierarchical clustering, the early estimation was finally improved [34].

$$BIC(J) = -2\sum_{j=1}^{J} \xi_j + m_J log(N) \qquad (2)$$

$$AIC(J) = -2\sum_{j=1}^{J} \xi_j + 2m_J \qquad (3)$$

where

$$m_J = J\left(2K^A + \sum_{k=1}^{K^B}(L_k - 1)\right) \tag{4}$$

In the final step, statistics such as the chi-square test for categorical variables and the t-statistics for numerical ones were utilized to evaluate the corresponding involvement of each variable in the formation of a cluster.

### 2.3. Association Rule Mining (ARM)

The ARM machine learning method is built on rules that allow us to identify meaningful rules from a huge dataset's many different properties. There is no requirement for predetermination in the ARM's hypotheses or forms of function [35]. In the traffic accident dataset, the different attributes that were accountable for an accident happening were determined by an association rule. Weka software (version 3.8.5) was used to apply the Apriori algorithm in order to find the useful rules [36].

Assuming a set of data $D$ with n occurrences, in which $T \in D$ for every transaction. Let $I$ be a set of elements, where $I = \{I_1, I_2, \ldots, I_m\}$. If $X \subseteq T$, then an item set $X$ will happen in $T$. If $X \subset I$, $Y \subset I$, and $X \cap Y = \emptyset$, then $X \rightarrow Y$ is an association rule. An implication of the form $X \rightarrow Y$, where $X$ is the antecedent and $Y$ is the consequent, is known as an association rule.

The following are some of the significance indicators used to assess a rule's quality in the ARM [16]:

Support ($S_p$) represents the frequency with which the item set occurs within the dataset. The support of a rule $X \rightarrow Y$ was computed as follows [37]:

$$Support\ \{X \rightarrow Y\} = P(X \cup Y) \tag{5}$$

Confidence ($C_f$) reflects the frequency with which the rule was confirmed to be accurate. The calculation of a rule $X \rightarrow Y$'s confidence was as follows [37]:

$$Confidence\ \{X \rightarrow Y\} = P(Y \mid X) = \frac{P(X \cup Y)}{P(X)} \tag{6}$$

Lift ($L_t$) was computed in this way [37]:

$$Lift\ \{X \rightarrow Y\} = \frac{P(Y \mid X)}{P(Y)} = \frac{P(X \cup Y)}{P(X)P(Y)} \tag{7}$$

if $L_t > 1$, this indicates that two occurrences were reliant on one another. These principles were helpful in forecasting the consequences in upcoming datasets.

Leverage ($L_v$) of rule $X \rightarrow Y$ was computed as follows [35]:

$$Leverage\ \{X \rightarrow Y\} = P(X \cup Y) - P(X)P(Y) \tag{8}$$

Conviction ($C_v$) of a rule $X \rightarrow Y$ was computed as follows [38]:

$$Conviction\ \{X \rightarrow Y\} = \frac{1 - P(Y)}{1 - P(Y \mid X)} \tag{9}$$

### 2.4. Kernel Density Estimation (KDE)

We can fully comprehend the point model's geographic change because of a variety of spatial analysis methods. After the accident data set was segmented into homogenous groups, the next important work was to locate the high-frequency accident locations. This enables traffic managers to take timely remedial measures. The accident hotspots were determined by using the GIS-based KDE technique [39–41]. This method involved placing a circular study area on every accident event using a kernel function, resulting in a smooth surface. The study area was then covered by a cell network. Next, a kernel function

was applied, which ranged from 1 at the event's center to 0 at the study area's radius (Figure 2) [29]. A spot's density was then calculated as follows [42]:

$$f(s) = \sum_{i=1}^{n} \frac{1}{\pi r^2} k\left(\frac{d_{is}}{r}\right)$$

(10)

where $f(s)$ is a position's density $s$, study radius is $r$, kernel function is $k$, and distance between $s$ and $i^{th}$ is $d_{is}$.

The KDE technique's result is displayed in raster format, which consists of a grid of cells. According to several studies, choosing the bandwidth $r$ is more crucial than choosing the kernel function $k$ [43]. There are three often used kernel functions: Minimum variance, Quartic, and Gaussian [44]. In this study, the Gaussian function was chosen and represented as:

$$k\left(\frac{d_{is}}{r}\right) = \frac{1}{\sqrt{2\pi}} exp\left(-\frac{d_{is}^2}{2r^2}\right), \; when \; 0 < d_{is} \leq r$$

(11)

$$k\left(\frac{d_{is}}{r}\right) = 0, \; when \; d_{is} > r$$

(12)

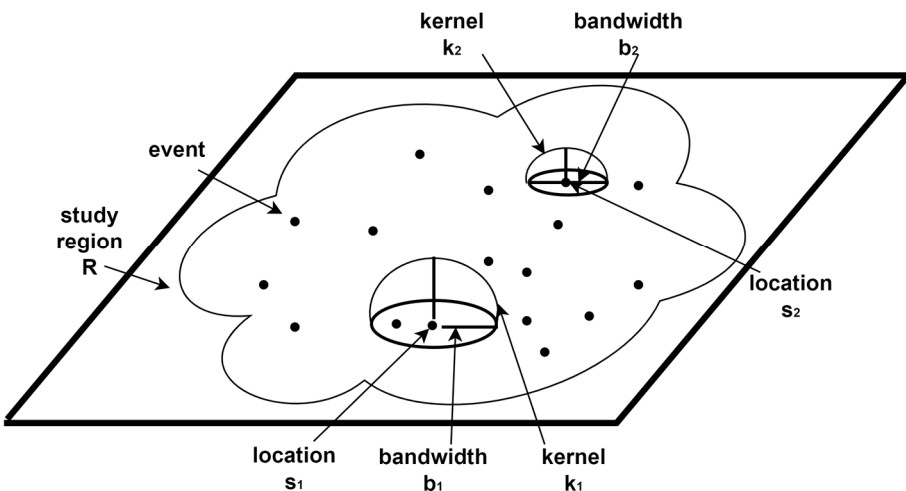

**Figure 2.** The figure illustrates the working principle of the kernel density technique [45].

## 3. A Case Study Analysis

### 3.1. Data and Study Field

#### 3.1.1. Field of Study

This study was carried out in Vietnam's capital, Hanoi. As of 2022, Hanoi has an area of 3328.9 km$^2$ and a population of 8.4 million people. Hanoi's main modes of travel are motorcycles, cabs, buses, and a growing number of personal vehicles. With more than 6.6 million cars on the road and more than 30,000 registrations added monthly, Vietnam's greatest percentage of personal vehicle growth is found in Hanoi. The Hanoi transportation infrastructure system is not keeping up with the current rate of urbanization, posing a significant risk of traffic accidents. Meanwhile, the analysis of the main factors causing traffic accidents has not yet been considered. Therefore, the author decided to choose Hanoi as a case investigation to test the suggested methodology.

#### 3.1.2. Research Data

A traffic accident dataset over three years (2015–2017) was investigated in Hanoi. According to many past studies, traffic accident (TA) data within 3 years is appropriate for analysis [15]. This dataset contained both categorical and numerical data. The significant information contained by this dataset encompasses the following: date, location, hour, and kind of accident, vehicle type, driver, wounded party details, etc.

### 3.2. Analysis Results and Discussions

3.2.1. Cluster Analysis

The procedure made use of every variable listed in Table 1. The BIC or AIC indicators were used to determine the cluster's number. The identical results were calculated through the parameters AIC and BIC. Table 2 shows the five groups with the highest distance measurement rate of 2.166. Therefore, five groups were selected by chance. However, the authors reran the model using the cluster's number set to 4, 6, and 7, correspondingly, to attempt to assess the group's number based on Figure 3. The ideal outcome was five groups in the end.

**Table 2.** Cluster selection criteria.

| Number of Clusters | AIC | Change in AIC | AIC Change Ratio | Distance Measurements Ratio |
|---|---|---|---|---|
| 1 | 72,941.018 | | | |
| 2 | 67,126.050 | −5814.969 | 1.000 | 1.893 |
| 3 | 64,209.883 | −2916.167 | 0.501 | 1.029 |
| 4 | 61,386.215 | −2823.668 | 0.486 | 1.277 |
| 5 | 59,247.404 | −2138.812 | 0.368 | 2.166 |
| 6 | 58,437.345 | −810.059 | 0.139 | 1.640 |
| 7 | 58,072.333 | −365.012 | 0.063 | 1.047 |
| 8 | 57,738.775 | −333.558 | 0.057 | 1.117 |
| 9 | 57,474.816 | −263.960 | 0.045 | 1.132 |
| 10 | 57,280.287 | −194.529 | 0.033 | 1.012 |

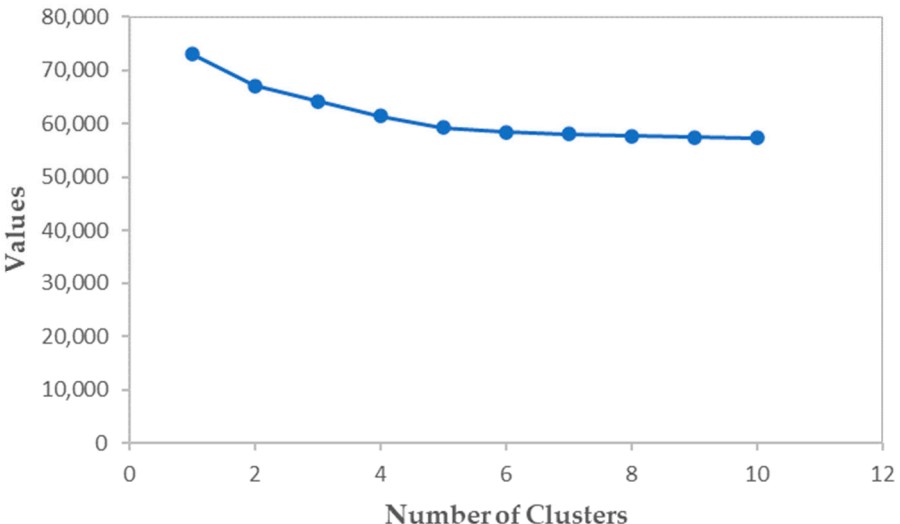

**Figure 3.** The selection criteria of cluster (AIC).

The distributions of cluster-dependent univariates for each variable were given in the five-cluster model. Thus, each cluster was determined as a specific TA type. The skewed feature distributions that vary among the clusters were considered to depict each cluster succinctly. For instance, if one cluster comprised more than 90% single-vehicle crashes while the others occupied similar low percentages for the feature "single-vehicle crashes", this cluster was considered as the "single-vehicle accident" cluster. The probability of the features enables us to assign the TA to different clusters. Table 3 shows the features and their probability in each cluster used to shape the five-cluster model.

**Table 3.** The probability of features in each cluster.

| Variable-Value | Group 1 (%) | Group 2 (%) | Group 3 (%) | Group 4 (%) | Group 5 (%) |
|---|---|---|---|---|---|
| Road type: national, provincial road | 10 | 90 | 0 | 40 | 75 |
| Road type: country lane | 0 | 2 | 92 | 20 | 5 |
| Road type: local street | 90 | 8 | 8 | 40 | 20 |
| The first user's [1] kind of vehicle: truck, car | 80 | 60 | 5 | 0 | 5 |
| The first user's kind of vehicle: motorbike | 10 | 30 | 86 | 93 | 82 |
| The presence of the second user | 100 | 100 | 100 | 0 | 100 |
| Vehicle type of the second user [2]: motorbike | 80 | 80 | 89 | 0 | 20 |
| Vehicle type of the second user: truck, car | 0 | 5 | 3 | 0 | 70 |
| Status of the first user: Fatal | 1 | 2 | 36 | 86 | 90 |
| Status of the second user: Fatal | 90 | 87 | 63 | 0 | 2 |

[1] The first user: the vehicle causes an accident; [2] The second user: the vehicles involved.

In cluster 1, 80% of vehicle kinds for the first road users were trucks and cars while 80% of vehicle kinds for the second traffic users were motorbikes. In total, 90% of all TA occurred on local streets. Therefore, cluster 1 was named as "TA between a truck/car and a motorbike on local streets".

Cluster 2 overlapped with cluster 1 because the first user's kind of vehicle still was predominated by truck and car (60%) and the second user's kind of vehicle was predominated by motorbike (80%). However, Table 3 discovered that 90% of all TAs in cluster 2 occurred on national and provincial roads. Thus, this cluster was depicted as "TA between a truck/car and a motorbike on national and provincial roads".

Cluster 3 was distinguished from other clusters by road type, which was a country lane in 92% of all cases. In addition, the vehicles of both road users in this cluster were motorbikes in around 86% and 89% of all cases. Thus, this cluster was depicted as "TA between two motorbikes on the country lanes".

In cluster 4, 100% of all accidents in this cluster were caused by themselves, without any other means. Table 3 found that this cluster was not like the rest of the groups because motorcycles were the user's vehicle in 93% of the cases. The authors refer to this cluster as "Single-vehicle motorbike crashes".

In cluster 5, the vehicle that caused the accident (illegal driving) was a motorbike (82%) while the second vehicle was legal driving. Thus, this cluster was depicted as "Motorbikes causing accidents on streets, provincial, and national roads".

Table 4 illustrates TA types corresponding to each cluster and their sizes. In addition, the outputs confirmed that the vehicle type and road type should be considered in segmenting TA data. Especially, this study determined a particular type of accident which was a motorbike accident caused by themselves (single-vehicle motorbike crashes). This is also one of the popular accident types in Vietnam, especially in big cities like Hanoi.

3.2.2. Association Rule Mining

The Apriori algorithm was used to create interesting rules. First, to generate useful rules, all of the frequent item sets in the dataset were identified by using a minimum support of 10%. After that, the useful rules were established by using these frequent item sets and the minimum confidence constraint (90%). It is important to identify the optimum support and confidence values in forming ARM.

The correlation among different attribute values occurring together when an accident happens was highlighted through association rules. There were many rules created but only several useful rules were selected based on lift value ($L_t > 1$). Table 5 shows the top ten best rules created for the whole dataset.

**Table 4.** Types of accidents.

| Cluster | TA Type | Size (%) |
|---|---|---|
| 1 | TA between a truck/car and a motorbike on local streets | 22 |
| 2 | TA between a truck/car and a motorbike on national and provincial roads | 27.8 |
| 3 | TA between two motorbikes on the country lanes | 12.3 |
| 4 | Single-vehicle motorbike crashes | 8.8 |
| 5 | Motorbikes causing accidents on streets, provincial, and national roads | 29.2 |

**Table 5.** Top ten best rules for the entire dataset.

| No | Best Rules | $C_f$ | $L_t$ | $L_v$ | $C_v$ |
|---|---|---|---|---|---|
| 1 | Speed limit = 80 km/h, first user = truck, second user = fatal → First user = no injuries | 0.99 | 2.12 | 0.06 | 37.2 |
| 2 | Single-vehicle crash = motorbike, over-speed → Fatal | 0.94 | 1.76 | 0.05 | 7.2 |
| 3 | Sparse area, speed limit = 60, first user = motorbike → Fatal | 0.91 | 1.69 | 0.05 | 4.6 |
| 4 | Dense area, Single-vehicle crash = motorbike → Fatal | 0.9 | 1.68 | 0.06 | 4.46 |
| 5 | Sparse area, first user = truck, second user = motorbike → Second user = fatal | 0.9 | 1.66 | 0.05 | 4.39 |
| 6 | Not paying attention, second user = fatal → First user = No injuries | 0.9 | 1.62 | 0.06 | 4.01 |
| 7 | Consequence 1 = No injuries, Status 2 = Fatal → Gender 1 = Male | 0.97 | 1.05 | 0.02 | 2.31 |
| 8 | Over-speed, Intersection → SI = Very severe | 0.96 | 1.05 | 0.02 | 2.09 |
| 9 | Speed limit = 80, Status 2 = Fatal → Gender 1 = Male | 0.96 | 1.04 | 0.02 | 1.93 |
| 10 | Reason = Over-speed, wrong lane → Gender 1 = Male | 0.95 | 1.03 | 0.01 | 1.36 |

Table 5 shows that single-vehicle crashes often occurred with motorbikes owing to over-speed in densely populated areas. Most of these accidents were often very serious. The road users who caused the accidents were often male. There were several main reasons such as over-speed, not paying attention, and wrong lane. On national and provincial roads, serious accidents often occurred between trucks and motorbikes in sparse areas. The result was that motorbike users were often fatal. However, it was very difficult to identify the main causative reasons if the dataset was not segmented into homogeneous clusters. Therefore, the application of ARM to each type of accident easily identified the main causes for each type. Table 6 depicts the association rules created for five clusters.

These rules are discussed as follows:

1. Rules for cluster 1

The robust rules identified a notable pattern: accidents frequently occurred at night in Cluster 1. The accident hotspots in this cluster were often intersections in local streets and densely populated areas, aligning with the findings from the previous step of accident classification. Two main types of vehicles dominated this cluster: trucks and motorbikes. The prevalent accident types included rear-end accidents and sideswipes. The primary causes of these accidents were determined to be unsafe following distance and illegal turns, identified by robust rules with high lift values.

The severity index for these accidents was predominantly serious and fatal. The first drivers involved in these accidents were often in the age range of 30–39. The scenarios in this cluster were frequently observed in Hanoi and other major Vietnamese cities. One common situation involved vehicles waiting at a red light while the vehicles behind, usually cars and trucks, were rushing forward. These instances often resulted in catastrophic outcomes.

2. Rules for cluster 2

The robust rules revealed that accidents in cluster 2 frequently occurred in the afternoon on provincial and national roads, as well as in spare areas. This observation aligns with the findings from the previous step of accident classification. The predominant type of accidents in this cluster was head-on accidents. The primary causes identified were illegal overtaking and driving in the wrong lane, which was determined by robust rules with high lift values.

**Table 6.** The best rules for five clusters.

| C * | No | Best Rules | $C_f$ | $L_t$ | $L_v$ | $C_v$ |
|---|---|---|---|---|---|---|
| 1 | 1 | Hour = Night, Reason = Unsafe distance → Kind = Rear-end | 1 | 3.16 | 0.1 | 8.2 |
| | 2 | Populated area = Yes, Reason = Unsafe distance → SI = Severe | 1 | 1.39 | 0.04 | 3.34 |
| | 3 | Reason = Unsafe distance → Kind = Rear-end, SI = Severe | 1 | 3.95 | 0.11 | 8. 96 |
| | 4 | Kind = Sideswipe, Intersection → Reason = Turning illegally | 1 | 6.08 | 0.11 | 8.35 |
| | 5 | Reason = Turning illegally, Intersection → Kind = Sideswipe | 1 | 6.58 | 0.11 | 8.48 |
| | 6 | Reason = Turning illegally, SI = Severe → Kind = Sideswipe | 1 | 6.58 | 0.1 | 7.63 |
| | 7 | Age 1 = 30–39, Kind = Sideswipe, SI = Severe, Intersection → Reason = Turning illegally | 1 | 6.08 | 0.08 | 6.68 |
| | 8 | Reason = Turning illegally, SI = Severe, Intersection → Kind = Sideswipe | 1 | 6.58 | 0.09 | 6.78 |
| 2 | 1 | Hour = Afternoon, Road type = Provincial road → SI = Severe | 0.93 | 1.22 | 0.02 | 1.67 |
| | 2 | Reason = Overtake illegally → SI = Severe | 0.92 | 1.21 | 0.02 | 1.55 |
| | 3 | Hour = Afternoon, Kind = Head-on → SI = Severe | 0.92 | 1.21 | 0.02 | 1.55 |
| | 4 | Road type = Provincial road, Age 1 = 30–39 → SI = Severe | 0.92 | 1.21 | 0.02 | 1.55 |
| | 5 | SI = Very Severe, Age 1 = 30–39 → Road type = National road | 0.92 | 1.37 | 0.03 | 1.98 |
| | 6 | Populated area = No, Reason = Wrong lane, Kind = Head-on, Road type = Provincial road → SI = Severe | 0.92 | 1.2 | 0.02 | 1.43 |
| | 7 | Reason = Wrong lane, SI = Severe, Age 1 = 30–39 → Kind = Head-on | 0.92 | 2.56 | 0.06 | 3.85 |
| | 8 | Reason = Wrong lane, Kind = Head-on, Age 1 = 30–39 → SI = Severe | 0.92 | 1.2 | 0.02 | 1.43 |
| 3 | 1 | Hour = Evening, Reason = Wrong lane, SI = Very Severe → Kind = Head-on | 1 | 1.36 | 0.12 | 5.6 |
| | 2 | Reason = Wrong lane, SI = Very Severe, Road type = Country lane → Kind = Head-on | 1 | 1.36 | 0.12 | 5.6 |
| | 3 | Age 1 = 24–29, Reason = Over-speed → SI = Very Severe | 1 | 1.36 | 0.05 | 2.4 |
| | 4 | Reason = Wrong lane, SI = Very Severe → Kind = Head-on | 1 | 1.36 | 0.12 | 5.6 |
| | 5 | Reason = Wrong lane, SI = Very Severe → Kind = Head-on, Road type = Country lane | 1 | 1.36 | 0.12 | 5.6 |
| | 6 | Intersection, Kind = Head-on, SI = Severe → Reason = Wrong lane | 1 | 1.5 | 0.04 | 2 |
| | 7 | Reason = Overs-peed, Kind = Head-on → SI = Very Severe | 1 | 1.36 | 0.04 | 1.6 |
| | 8 | Populated area = No, Reason = Wrong lane → Kind = Head-on | 0.9 | 1.23 | 0.11 | 2 |
| 4 | 1 | Populated area = Yes, Kind = Out-of-control, SI = Severe → Reason = Over-speed | 1 | 1.05 | 0.02 | 1.26 |
| | 2 | Hour = Night, SI = Severe → Reason = Over-speed | 1 | 1.05 | 0.01 | 0.91 |
| | 3 | SI = Severe → Reason = Over-speed | 0.98 | 1.02 | 0.02 | 1.04 |
| | 4 | Hour = Night → Reason = Over-speed | 0.97 | 1.01 | 0.01 | 0.7 |
| | 5 | Dense area, Age 1 = 24–29 → Reason = Over-speed | 0.96 | 1.01 | 0 | 0.59 |
| | 6 | Hour = Evening → Reason = Over-speed | 1 | 1.05 | 0.01 | 0.61 |
| | 7 | Hour = Night, Age 1 = 24–29 → Reason = Over-speed | 1 | 1.05 | 0.01 | 0.7 |
| | 8 | SI = Severe, Age 1 = 24–29 → Reason = Over-speed | 1 | 1.05 | 0.01 | 0.7 |
| 5 | 1 | Hour = Afternoon, Kind = Head-on, SI = Severe, Road type = National road → Reason = Wrong lane | 0.9 | 2.51 | 0.08 | 4.48 |
| | 2 | Age 1 = 30–39, Reason = Wrong lane, SI = Severe, Road type = National road → Kind = Head-on | 0.9 | 2.89 | 0.08 | 4.81 |
| | 3 | Age 1 = 30–39, Reason = Wrong lane, Road type = National road → Populated area = No | 0.97 | 1.56 | 0.07 | 5.51 |
| | 4 | Reason = Wrong lane, Road type = National road, Speed limit = 80 → Populated area = No | 0.96 | 1.55 | 0.06 | 4.94 |
| | 5 | Reason = Wrong lane, Kind = Head-on, Road type = National road → Populated area = No | 0.96 | 1.55 | 0.05 | 4.56 |
| | 6 | Intersection, Kind = Head-on, Road type = National road, Speed limit = 80, Populated area = No → Reason = Wrong lane | 0.96 | 2.66 | 0.09 | 7.36 |
| | 7 | Kind = Head-on, Road type = National road → Populated area = No | 0.93 | 1.5 | 0.06 | 3.67 |
| | 8 | Hour = Night, Populated area = Yes → SI = Severe | 0.96 | 1.22 | 0.03 | 2.45 |

* C = Cluster.

These accidents were predominantly classified as serious or fatal, as indicated by the severity index. The first drivers involved in these accidents were often in the age range of 30–39. Many provincial and national roads in this area lacked a center divider and had a narrow width. Consequently, numerous vehicles engaged in illegal overtaking into the opposite lane, leading to catastrophic head-on accidents.

3.    Rules for cluster 3

The robust rules revealed that accidents in cluster 3 frequently occurred in the evening, primarily at intersections in country lanes and spare areas. The predominant type of accidents in this cluster was head-on accidents. The main causes identified were driving in the wrong lane and over-speeding, as determined by robust rules with high lift values.

The first drivers involved in these accidents were often in the age range of 24–29. The severity index for these accidents was primarily categorized as very severe. Similar to cluster 2, many country lanes in this area lacked a center divider and had a narrow width. Consequently, numerous vehicles engaged in illegal overtaking into the opposite lane, leading to catastrophic head-on accidents.

4.    Rules for cluster 4

The robust rules revealed that accidents in cluster 4 frequently occurred in dense areas during the evening and at night. Motorbikes were the primary types of vehicles involved in this cluster. The major type of accident was loss of control leading to solo crashes. The main causes identified for these accidents were over-speeding, out of control, and limited visibility at night, all determined by robust rules with high lift values.

The severity index for these accidents was primarily categorized as severe. This category of accidents typically involved male drivers aged 24–29.

5.    Rules for cluster 5

The robust rules revealed that accident locations in cluster 5 frequently occurred at intersections during the afternoon and at night, in both sparse and dense areas. Motorbikes were the first road users responsible for causing accidents in this cluster. The major types of accidents were head-on accidents. The main causes identified for these accidents were driving in the wrong lane and exceeding the speed limit, as determined by robust rules with high lift values.

This category of accidents typically involved male drivers aged 30–39. The severity index for these accidents was primarily categorized as severe.

### 3.2.3. Determination of Hotspots in Each Cluster

Currently, there are not many countries that identify accident hotspots based on the repeatability criteria of the same type of accident or cause. In our opinion, a location that has the repeatability of the same type of accident or cause should be considered an important hotspot that needs to be dealt with promptly. In this study, after identifying the specific types of accidents and their associated causes, the next task was to identify hotspot locations in each of these clusters. Based on the proposed method, the process of identifying hotspots becomes convenient and accurate. Figure 4 shows that the red color areas were the hotspots that were determined based on the repeatability of the same type of accident or cause.

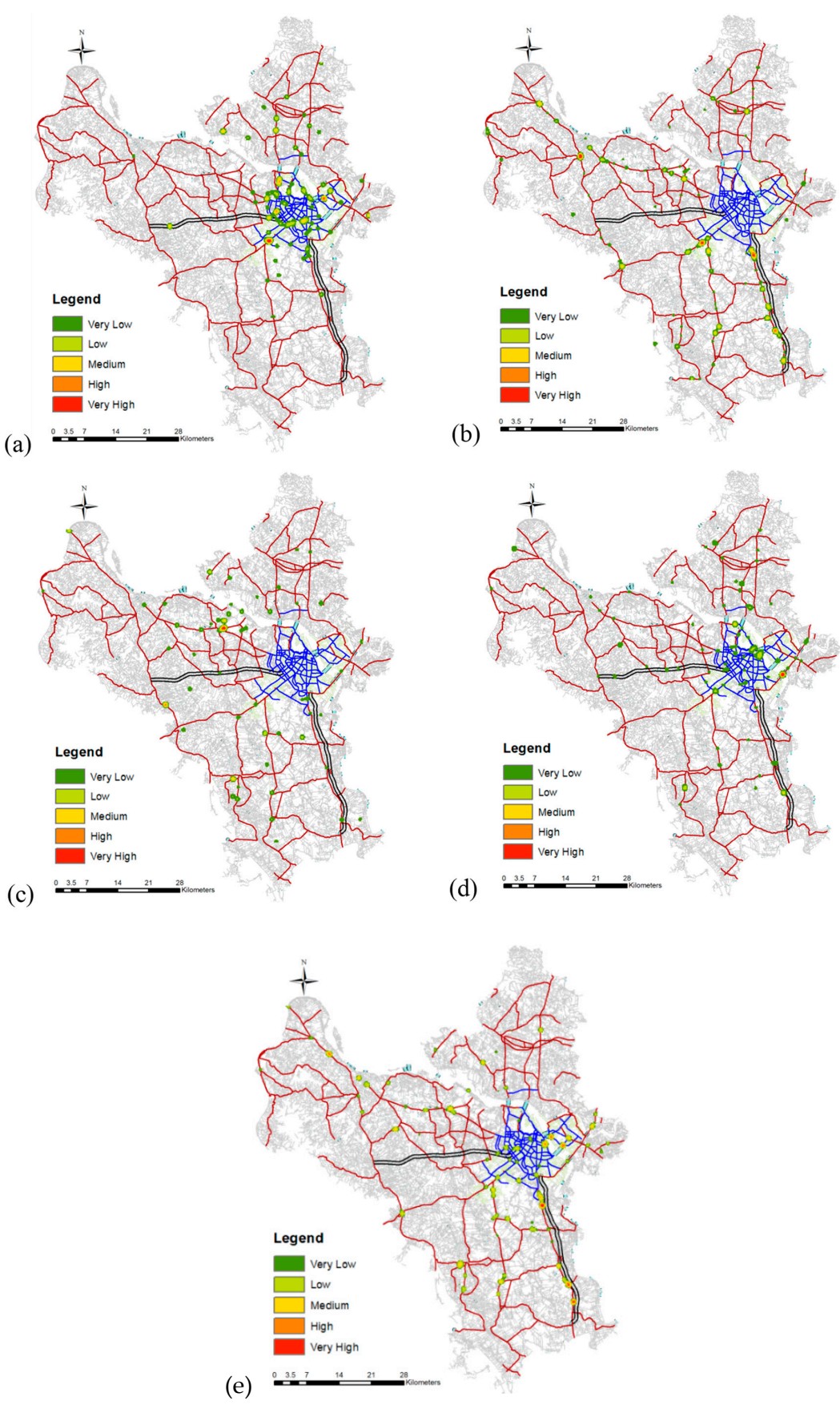

**Figure 4.** Hotspots in clusters (blue lines are roads in the city center). (**a**) Cluster 1; (**b**) Cluster 2; (**c**) Cluster 3; (**d**) Cluster 4; (**e**) Cluster 5.

Table 7 shows a summary of the types of accidents, main causes, and recommendations for remedies.

**Table 7.** TA types, main reasons, and preventive strategies.

| C * | TA Type | TA Kind | Main Reasons | Time | Area | Suggestion |
|---|---|---|---|---|---|---|
| 1 | TA between a truck/car and a motorbike on local streets | Sideswipe, Rear-end | Unsafe distance, turning illegally, (Age1 = 30–39) | Night | Densely | Law enforcement, education, engineering improvement. |
| 2 | TA between a truck/car and a motorbike on national and provincial roads | Head-on | Overtake illegally, wrong lane (Age1 = 30–39) | Afternoon | Sparely | Removing illegal intersections or adding the roadway with side security precautions including pedestrian crossings, median strips, lighting, and speed limit signs. |
| 3 | TA between two motorbikes on the country lanes | Head-on | Wrong lane, Over-speed (Age1 = 24–29) | Evening | Sparely | Law enforcement, education, engineering improvement, improving road surface. |
| 4 | Single-vehicle motorbike crashes | Out of control | Over-speed, out of control, limited visibility (Age1 = 24–29) | Evening, night | Densely | Law enforcement, education, engineering improvement. |
| 5 | Motorbikes causing accidents on streets, provincial, and national roads | Head-on | Wrong lane, high-speed limit (Age1 = 30–39) | Afternoon, night | Sparely, Densely | Law enforcement, speed limit signs. Removing illegal intersections or adding the roadway with side security precautions including pedestrian crossings, median strips, lighting, and speed limit signs. |

* C = Cluster.

## 4. Validation of the Results

Correlation analysis was applied to validate the outputs of the proposed method. Importantly, correlation analysis was carried out between severity index (SI) and various factors such as time intervals of the day, reasons, type of road, kind of accident, age, gender, and vehicle type of driver.

The findings in the clusters indicated that trucks/cars and motorbikes were the main vehicles involved in serious accidents. The findings in the clusters also indicated that head-on, rear-end, and angle were the main kinds of accidents. In addition, the findings showed that over-speed, wrong lane and not paying attention were the main reasons causing accidents. All of these findings are in accordance with the outcomes presented in Table 8. Moreover, Table 8 also shows that the age of the first road users, type of the second vehicle, time, speed limit, and surroundings had a significant relationship to the severity level of accidents. Thus, the results from the proposed method satisfied the validation process. The gained results from the proposed method were reliable and exact.

**Table 8.** Correlation analysis between SI and contributing factors.

| Factors | Chi-Square Tests | Outputs |
|---|---|---|
| SI and time intervals of the day | $\chi^2 = 23.938$ $p < 0.05$ | The test is significant. The test indicates that there is a significant relationship between SI and time intervals of day. Severe crashes often occurred afternoon (8.9%), evening (14.9%), and night (5.8%) while morning (5.0%). |

| Factors | Chi-Square Tests | Outputs |
|---|---|---|
| SI and the age of the first user | $\chi^2 = 27.598$ $p < 0.05$ | The test is significant. The test indicates that there is a significant relationship between SI and the age of the first user. Severe crashes often occurred in groups 24–29 (9.5%) and 30–39 (10.5%), higher than others. |
| SI and reasons | $\chi^2 = 37.391$ $p < 0.05$ | The test is significant. The test indicates that there is a significant relationship between SI and reasons. Severe crashes often occurred in accordance with over-speed (12.1%), wrong lane (9.3%), no paying attention (6.2%), higher than others. |
| SI and the vehicle of the first user | $\chi^2 = 55.539$ $p < 0.05$ | The test is significant. The test indicates that there is a significant relationship between SI and the vehicle of the first user. Severe crashes often occurred in accordance with motorbikes (30%) trucks (23.3%), and cars (8.7%), higher than others. |
| SI and the vehicle of the second user | $\chi^2 = 31.091$ $p < 0.05$ | The test is significant. Severe crashes often occurred in accordance with motorbikes (27.7%) and trucks (14.9%), higher than others. |
| SI and accident-type | $\chi^2 = 17.247$ $p < 0.05$ | The test is significant. Severe crashes often occurred in accordance with head-on (10%), angle (7.3%), rear-end (7.3%), higher than others. |
| SI and populated area | $\chi^2 = 9.379$ $p < 0.05$ | The test is significant. Severe crashes often occurred in accordance with sparely populated areas (suburbs) (22.4%), higher than others. |
| SI and speed limit | $\chi^2 = 8.894$ $p < 0.05$ | The test is significant. Severe crashes often occurred at higher speed limits (25.8%). |

## 5. Conclusions, Limitations, Suggestions, and Future Work

### 5.1. Conclusions

This study proposed the integration of GIS and data mining techniques in TA analysis. The two-step algorithm applied in data segmentation brought good results. This algorithm overcomes the drawbacks of methods like K-means, K-Modes, K-Medoids, and LCC because it not only handles both types of data including numerical and categorical data, but it also determines the optimal number of clusters automatically. This algorithm determined five common types of accidents in Hanoi during the research period. In detail, cluster 1 was accidents between a truck/car (80%) and a motorbike (80%) on local streets, accounting for 22%. Cluster 2 presented accidents between a truck/car (60%) and a motorbike (80%) on national and provincial roads, accounting for 27.8%. Cluster 3 illustrated accidents between two motorbikes (86% and 89%) on the country lanes (92%), accounting for 12.3%. Cluster 4 depicted single-vehicle motorbike crashes (100%), with the lowest rate of 8.8% within the groups. Finally, cluster 5 depicted accidents caused by motorbikes (82%) which made up the highest percentage within the groups, accounting for 29.2%. In addition, the locations of the types of accidents were represented visually on a map. This enables traffic authorities to propose accurate and urgent countermeasures.

This study also confirmed that the TA dataset was grouped into homogeneous clusters that facilitated the identification of the causes more easily and accurately. It was difficult to identify the main causative reasons if the dataset was not segmented into homogeneous clusters. Moreover, the results of the study also identified the types of accidents, the main causes, the time as well as the surrounding areas corresponding to each accident group. In detail, unsafe distance keeping and illegal crossing in densely populated areas at night were the main factors causing accidents in cluster 1 ($C_f = 1$ and $L_t \geq 1.39$). For cluster 2, illegally overtaking and driving in the wrong lane in sparsely populated areas in the afternoon were the main causes of accidents ($C_f \geq 0.92$ and $L_t > 1.2$). For cluster 3, speeding and driving in the wrong lane in the evening in sparsely populated areas were the main causes ($C_f \geq 0.9$ and $L_t \geq 1.23$). For cluster 4, speeding, out of control, and limited visibility in the evening in densely populated areas were the main causes ($C_f \geq 0.96$ and $L_t \geq 1.01$). For cluster 5,

speeding and driving in the wrong lane in the afternoon and at night were the main causes in this cluster ($C_f \geq 0.9$ and $L_t \geq 1.22$).

The accident causes were analyzed in detail in each cluster. In general, the results showed that motorbikes were the vehicles that often caused accidents. The first road user that caused accidents was often male aged from 24 to 39 years old. Common types of accidents were rear-ended, head-on, and out of control. The main causes were over-speed, wrong lane, overtaking illegally, and out of control. Analyzing the causes of accidents for each cluster enables traffic authorities to understand the reasons behind each accident and to take appropriate remedies.

Importantly, currently, there are not many countries that identify accident hotspots based on the repeatability criteria of the same type of accident or cause. In our opinion, a location that has the repeatability of the same type of accident or cause should be considered an important hotspot that needs to be dealt with promptly. In this suggested method, after identifying the specific types of accidents and their associated causes, the next task was to identify hotspot locations in each of these clusters. Based on the proposed method, the process of identifying hotspots becomes convenient and accurate. Thus, my proposed methodology can deal with the shortages from the previous studies. This enables traffic authorities to propose accurate and urgent countermeasures. The findings indicate an opportunity for many areas, particularly traffic accidents, to incorporate GIS and data mining approaches.

### 5.2. Limitations

The limitation of this research was the scope and scale of the data and study area because collecting traffic accident data in Vietnam faces many difficulties and challenges. If the data set is collected over a longer period of time and the research area is expanded to include other cities, the results will more fully and accurately reflect traffic accident situations in Vietnam. However, the authors still hope that the results of this research will help authorities understand in more detail the typical types of traffic accidents as well as the important factors causing serious accidents. Additionally, the methods employed could potentially be applied to other cities, providing an added avenue for analysis.

### 5.3. Suggestions

First, this study suggests that the TA dataset should be grouped into homogeneous clusters that will facilitate the identification of the causes more easily and accurately. Second, the two-step algorithm applied in data segmentation brought good results. This algorithm overcomes the drawbacks of methods like K-means, K-Modes, K-Medoids, and LCC because it not only handles both types of data including numerical and categorical data, but also determines the optimal number of clusters automatically. Third, the results also confirmed that the type of vehicles, type of road, and accident types can be applied to segment the data. Next, in our opinion, a location that has the repeatability of the same type of accident or cause should be considered as an important hotspot that needs to be dealt with promptly. Finally, the findings indicate an opportunity for many areas, particularly traffic accidents, to incorporate GIS and data mining approaches.

### 5.4. Future Works

In future research, we will expand the data scope and time span. We will analyze traffic accident data in more areas and over a longer period of time to obtain more comprehensive findings. Next, we will integrate traffic accident data with additional data sources, such as traffic flow data and weather data, to further investigate the contributing factors of traffic accidents. Also, the methods of cluster analysis and ARM need to be further optimized to improve the accuracy and stability of the model.

**Author Contributions:** Conceptualization, K.G.L.; Data curation, K.G.L. and Q.H.T.; Formal analysis, Q.H.T.; Methodology, K.G.L. and Q.H.T.; Supervision, K.G.L.; Validation, K.G.L.; Visualization, Q.H.T.

and V.M.D.; Writing—original draft, K.G.L.; Writing—review & editing, K.G.L. All authors have read and agreed to the published version of the manuscript.

**Funding:** This research is funded by University of Transport and Communications (UTC) under grant number T2022-CT-008TĐ.

**Data Availability Statement:** The data presented in this study are available on request from the corresponding author.

**Conflicts of Interest:** The authors declare no conflict of interest.

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
