# Peer review of "Urban Traffic Accident Features Investigation to Improve Urban Transportation Infrastructure Sustainability by Integrating GIS and Data Mining Techniques"

_sustainability, doi:10.3390/su16010107_

Round 1

Reviewer 1 Report

Comments and Suggestions for Authors

If Abstract summarizes the main results of the paper using quantitative numerical data, the specificity and reliability of the study will be improved.

Some modifications are required in the edit format (ex. Insert line between line 444 and line 445, before Section 4).

Section 4 also needs to summarize the key findings of the study using quantitative numerical data, which is likely to improve the reliability of the study and the reader's understanding.

It would be better to separate the Discussion section, and mention the implications and limitations of the study and the author's suggestions in the Discussion section, and to describe it separately from the Conclusion section would allow the reader to better understand the objective analysis results and the author's suggestions (or implications).

Author Response

Thank you very much for your comments and suggestions for our research. We have updated the details in the attached revised manuscript.

Reviewer 2 Report

Comments and Suggestions for Authors

The manuscript proposes a methodology for studying and charactering traffic accidents, in order to reduce them. The main types and causes of the accidents were identified, using data of three years from Hanoi.
The method used is based on Data Mining, instead of traditional statistical models. Several clustering techniques have been considered, which are not valid for this study, justifying the use of ARM, which has been previously used in other works.
Finally, in order to identify the location of the hotspots, the repeatability of same a type of accident was analised, and the data are graphically visualized by using GIS.

The manuscript is well written and structured. The methodology is properly described.

* About Acronyms, nearly of them are well defined at their first usage, except the following one: TA (line 281, although it is defined in the Keywords and line 198).

* Some minor typing mistakes or suggestions:
- line 400: "3.2.2.5 Association rules for cluster 5" -> "3.2.2.5 Rules for cluster 5"

* About tables/figures:
- pg.7: Figure 2 is a bit pixelated.
- pg.9: There is another Figure 2 (instead of Figure 3) and it is also pixelated. So, the current Figure 3 should be Figure 4, and also in Table 7 (F3a -> F4a, and so on). Perhaps, on Table 7, in order to avoid errors because of this, the column "H" (even "TA type") could be removed.
- pg.12-14: Table 6 is a very large and a bit confusing. Perhaps an horizontal line between clusters could improve the readability.

* About references/bibliography:
All the references seems to be appropriate and updated.
- In line 85, "Kumar and Toshniwal (2015)" is not referenced by its index [5].
- in [3], the link "https://bocongan.gov.vn/" is not accesible, at least from my location.
- In [13], a preprint from 2021 is included, but it seems that the final work is already published.
https://link.springer.com/chapter/10.1007/978-981-16-0586-4_12
- Different text fonts or sizes are used along the references.

Author Response

(The authors gave the same response as above.)

Reviewer 3 Report

Comments and Suggestions for Authors

1. The study only selected data from 2015 to 2017 as case studies, which may not fully represent the overall situation of traffic accidents.

2. In the data preparation stage, the processing methods for noise, missing values and outliers are not explained in detail, which may have a certain impact on the results.

3. The results of cluster analysis and association rule mining may be affected by parameter selection and need to be further optimized and verified.

Possible research directions (4-6):

4. The research can expand the data scope and time span, and analyze traffic accident data in more regions and over a longer period of time to obtain more comprehensive conclusions.

5. The research can be combined with other data sources, such as traffic flow data and weather data, to further explore the influencing factors of traffic accidents.

6. The method of cluster analysis and association rule mining can be further optimized to improve the accuracy and stability of the model.

Comments on the Quality of English Language

minor revision

Author Response

(The authors gave the same response as above.)

Round 2

Reviewer 3 Report

Comments and Suggestions for Authors

1. When submitting the article, check whether the format of the article is correct. There are some format problems in the article, such as the table name position "Table 1. The variables of traffic accident.".

2. In the association rule mining part, the article mentions the "ARM machine learning method", and there are many algorithms in the association rule mining. Can you explain in detail what the algorithms used in the traffic data set mining are? Why do you use this algorithm?

3. It is mentioned that "a two-step algorithm is applied in data segmentation, which brings a good response". Can you specifically compare the two algorithms, K-means, K-models, K-Medoids and LCC? Expansion illustrates the advantages of the two-step algorithm.

Comments on the Quality of English Language

Minor editing of English language required.

Author Response

(The authors gave the same response as above.)
